# Infection and Risk Perception of SARS-CoV-2 among Airport Workers: A Mixed Methods Study

**DOI:** 10.3390/ijerph17239002

**Published:** 2020-12-03

**Authors:** Jeadran Malagón-Rojas, Eliana L. Parra B, Marcela Mercado

**Affiliations:** 1Facultad de Medicina, Universidad El Bosque, Bogotá 110121, Colombia; 2Subdirección de Investigación Científica y Tecnológica, Instituto Nacional de Salud, Bogotá 111321, Colombia; elparra@ins.gov.co (E.L.P.B.); mmercado@ins.gov.co (M.M.)

**Keywords:** SARS-CoV-2, COVID-19, mixed methods, workers, occupational health, public health

## Abstract

This is a mixed-methods research study carried out on a cohort of airport workers during the SARS-CoV-2 pandemic. We used quantitative and qualitative methods to describe the infection and risk perception of SARS-CoV-2 in a cohort of workers at the International Airport El Dorado/Luis Carlos Galán Sarmiento in Bogotá, Colombia. An incidence of SARS-CoV-2 infection of 7.9% was found in the workers. A high perception of risk was associated with activities such as using public transport. Risk perception is strongly influenced by practices related to work conditions and environments. These findings could help us understand the pandemic’s dynamics and the conceptions of the risk of transmission to promote policies on health and safety in this group of workers.

## 1. Introduction

After the alarm generated by the appearance of an outbreak of atypical pneumonia in China in December 2019, the entire world has been working to implement measures to contain the spread of the SARS-CoV-2 virus [1]. These measures have included the closure of educational and commercial institutions, restrictions on mobility, and forced quarantine of the population [1]. However, personnel working in health services, food services, banking services, provision of public services, and transportation have continued to work [2,3].

In a globalized society, the transport sector is crucial for the development of the economy, politics, and other facets of society. It is estimated that before the pandemic, 9 million people traveled by air daily [4]. The mobility of passengers from different places around the world, who are confined in cabins and waiting rooms for long periods of time, represents a risk of increasing the transmission of the virus. In this vein, airports may require the development of surveillance systems that help prevent the spread of infections among passengers and workers, especially at international airports, where the largest number of travelers converge [5].

At the same time, the World Health Organization (WHO) has established a series of functions that airport workers perform that may increase person-to-person contact, such as measuring the body temperature of travelers and the use of questionnaires [6]. For these reasons, some of the activities carried out by airport workers have been designated as “high-risk duties”, equating them to the risks faced by workers in the health sector [7,8]. A risk level scale has been proposed by some authors to develop a score for risk occupations of COVID-19 infection to estimate the proportion of workers exposed to the virus based on factors such as physical proximity, contact with potential asymptomatic carriers, and work conditions [7,8]. Nevertheless, this approximation of the risk of transmission does not include the individual perception of the risk, which has been stated by various theorists of risk as a relevant element for the comprehension of behavior and risk [9,10]. From this perspective, perception should be understood as the knowledge, senses, and practices that determine how individuals are able to understand a given context and make decisions to protect themselves [11]. Hence, for a more holistic landscape of the risk of transmission of SARS-CoV-2, an approximation is required that may include sociodemographic elements related to individual differences (sex, age, and educational level, among others), clusters of analyses that involve the cognitive tradition (the level of knowledge and understanding of a given risk), and emotional and experiential tradition (personal experience) [12].

The role of airport terminals in the dissemination of the SARS-CoV-2 pandemic is well known [13,14,15]. The El Dorado International Airport/Luis Carlos Galán Sarmiento in Bogotá, Colombia, is no different. Epidemiological analyses have indicated that the first cases entered the country via El Dorado. This is understandable, considering that this terminal moves approximately 30 million passengers per year [16]. In addition, El Dorado is the third most important connection hub in the region [16]. Despite this, El Dorado was formally closed for commercial flights in late March. Nevertheless, since being quarantined at home was not an option for most workers because they had to keep working for cargo operations and “humanitarian trips”, we designed a mixed-methods study to understand the role of risk perception in the transmission of SARS-CoV-2 among the airport workers of these frontline airport services.

## 2. Materials and Methods

### 2.1. Study Design

We performed an explanatory sequential mixed study [17,18]. The quantitative component was a prospective cohort study, followed by the qualitative descriptive component [17,19]. The length of the follow-up was three months. Workers were followed every 21 days, from 1 June to 30 September 2020. The study was designed following the recommendations of the STROBE checklist and the SRQR statement [20] for observational studies and quality standards for reporting qualitative research results [21] (see Appendix A). The methods and processes are described in Figure 1.

### 2.2. Population

The study included a cohort of male and female workers between 18 and 60 years old with a current work contract involving face-to-face activities at the El Dorado International Airport between March and July 2020. The staff were from the areas of migration and/or emigration, reception, passenger care, and common area cleaning. At the beginning of the study, the population consisted of 500 workers distributed over three shifts. Workers who had contact with anyone infected with SARS-CoV-2 outside of work (relatives at home and extended family) and workers in telework mode were excluded.

### 2.3. Sample and Sampling

The sample size was calculated using the Ministry of Health estimates for the incidence of SARS-CoV-2 infection in the general population [22] using the OpenEpi^®^ statistical program [23]. A sample size of 205 workers, with an associated 95% CI, a precision of 2.3%, and a margin of error of 5%, was used. In order to determine the number of workers for each area, a proportional allocation was used (Appendix A).

### 2.4. Specific Hazard Assessment Matrix for SARS-CoV-2

The information on hazards in the workplace was extracted from the hazard identification matrix, with the sources of exposure, time, and type of exposure that workers may have (direct contact with drops or aerosols and/or indirect contact with contaminated surfaces).

### 2.5. The Sociodemographic Survey, Risk Perception Scale, and Epidemiological Files

A questionnaire was constructed to characterize occupational exposure to SARS-CoV-2, following the instruments recommended by the WHO [24]. The COVID-19 Risk Perception Questionnaire was also included [25]. The risk perception questionnaire uses six questions, where participants rate their agreement with the statements on a Likert-type scale from 1 to 10. The higher the score is, the greater the perception of risk of COVID-19 as a threat. The questions were the following: (i) How much does coronavirus infection affect your life? (ii) How long do you think the coronavirus infection will last? (iii) To what extent do you feel symptoms due to coronavirus infection? (iv) How concerned are you about coronavirus infection? (v) How emotionally does the coronavirus infection affect you? (vi) How exposed do you feel to the coronavirus at your workplace?

From both instruments, a single instrument was designed that grouped sociodemographic variables, presence or absence of symptoms, use of personal protection elements, occupational and extraoccupational practices, and perception of the threat of COVID-19. The document was subject to content validation [26] by three thematic experts in occupational health and biosafety. Additionally, the instrument was piloted on a group of workers from the National Institute of Health. Moreover, the epidemiological report from Acute Respiratory Infection by New Virus, code 346 of the Epidemiological Surveillance System SIVIGILA, was used.

### 2.6. Biological Samples

Nasopharyngeal samples from the workers were taken to determine COVID-19 viral RNA. The technique for collecting the nasopharyngeal swab sample is described in the guide for laboratory surveillance of the influenza virus and other respiratory viruses of the National Institute of Health [27]. The sample was taken by personnel trained in the technique. The detection of the virus was performed by RT-PCR according to the Berlin protocol previously described and standardized at the National Institute of Health [28].

The results were delivered to each worker through the company’s occupational medicine division. In the case of positive cases, the workers were contacted by telephone to report the results.

### 2.7. Worker Contacts

Each worker with a positive PCR result underwent a study of work and family contacts. Close contact was defined as someone who was within 2 m of the workers for a cumulative total of 15 min or more, starting from two days before the RT-PCR test (Figure 1). One of the research team members carried out a telephone call to characterize the contacts within the last two days. A checklist was used to document the fulfillment of close contact criteria. A nasopharyngeal swab sample was taken from each contact (Appendix A).

### 2.8. The Follow-Up of Cases at Home

The workers who tested positive for SARS-CoV-2 were followed up at home on the 7th, 14th, and 21st days after the first sample was taken. The follow-up at home was carried out by two researchers trained in taking samples and filling out the forms. At the follow-up visit, a new sample was taken to show when the test became negative, in addition to monitoring the health status of the worker and the study of contacts.

### 2.9. Semistructured Interviews

To develop the perceptions of risk and emotions generated by the SARS-CoV-2 pandemic in the group of workers who were infected [29], an interview guide was prepared by the principal investigator. The questions were formulated from a review of the theoretical referents around the issue of perception of health risk [11,25], health practices [30], and emotions against risk [31]. Validation was carried out with thematic experts (an expert in qualitative approach and an expert in occupational safety and health) [26]. Subsequently, the interview was piloted [32] on a group of healthy airport workers (*n* = 3).

### 2.10. Sample and Sampling

Snowball sampling was defined among workers who had positive results in the RT-PCR test. Workers with positive results were invited for the interview. The theoretical saturation criterion was considered to complete the sample size [33].

### 2.11. Interview Categories

The pre-established categories were formed based on a literature review performed by two of the researchers. The pre-established categories were validated by an expert on qualitative research. The included pre-established categories were (i) perception of risk in the face of the contagion of coronavirus at the workplace and at home; (ii) intra-and extraoccupational risk practices related to the spread of SARS-CoV-2, and (iii) emotions regarding the coronavirus. Based on the categories, the research team designed a semistructured interview guide, which is presented in Table 1.

### 2.12. Conducting the Interview

The interview was scheduled with each worker approximately 40 days after the COVID-19 diagnosis. Sessions were planned to last 60 min. All of the interviews were performed in Spanish. The interviews were recorded on a Sony Integrated USB Digital Voice Recorder-Icd-px470.

### 2.13. Information Processing and Analysis—Quantitative Data

The data were collected in an electronic GoogleForms^®^ survey, from which a Microsoft Excel^®^ V. 2019 database was generated. The statistical package Statistical Package for the Social Sciences (SPSS) V 22.0 (SPSS Inc., Chicago, The United States of America) (National Health Institute license) was used for the analysis.

For the quantitative variables, means and standard deviation were estimated. In the qualitative variables, frequencies and percentages were determined. Subsequently, a bivariate analysis was performed comparing the nominal or ordinal variables regarding the presence or absence of secondary infection by SARS-CoV-2, analyzed using Pearson’s chi-square test with Yates correction [34]. In the case of quantitative variables, the Spearman correlation was used [35]. The level of statistical significance established was *p* < 0.05.

A Poisson regression model was used to estimate the relative risk to evaluate the differences between groups [36]. In order to determine the variables for the model, we performed a literature review to identify risk factors for SARS-CoV-2 transmission. The included variables were related to sociodemographic characteristics, usage of protective elements, job position, comorbidities, transport mode, and conformation of the family. The adjusted relative risks (RR), with their respective 95% confidence intervals, were estimated to evaluate the differences between the groups using negative results as a reference. Additionally, accumulated incidence and incidence density (cases per 100 persons/month) were calculated. Cox regression was used to estimate the hazard function (per person/trimester).

### 2.14. Information Processing and Analysis—Qualitative Data

The processing and analysis of the qualitative data were guided using strategies to ensure credibility [37], trustworthiness, and transferability [38]. Credibility was achieved by following the semistructured interview guideline, followed by analytic triangulation performed by an expert not linked to the project [39]. To accomplish trustworthiness, the interviews were transcribed into Word^®^ and then converted into TXT format, removing accents and vignettes. Subsequently, the online platform QCA map was used (available at https://www.qcamap.org ) for the analysis of qualitative information. One of the researchers analyzed the transcripts independently. When 20% of the transcribed material was covered, the final analysis categories were established. Following this, the document was reviewed again to perform the categorization and axial coding. The codes and emerged categories were validated by a second researcher [40]. The findings were discussed by two of the coauthors until consensus on categories and subcategories was achieved. Finally, from the selective coding, a description of the content was made to determine the perceptions of the participants regarding the theoretical categories and to determine the relationships between the meanings of the categories [41]. The scope of the analysis was limited to a descriptive level of content.

Transferability was established by including sufficient quotations collected through in-depth interviews. Additionally, the audio and transcription files were stored in MP4 format and on an external hard drive to ensure that all of the phases of analysis could be traced back to original interviews. Additionally, the QCAmap^®^ (Verein zur Förderung qualitativer Forschung–Association for Supporting Qualitative Research ASQ, Wörthersee, Austria) generated a downloadable file with results that showed the frequency of the assigned codes in all the registration units.

### 2.15. Triangulation and Integration of Results

The study included data integration through the narrative in a contiguous approach [42]. First, a set of quantitative results was generated, followed by qualitative results, and, later in the discussion, the findings of each data set were integrated [43].

### 2.16. Ethical Considerations

This study considered the aspects indicated by the Declaration of Helsinki on research ethics (World Medical Association, 2013) and Resolution 8430 of 1993 of the Ministry of Health that establishes academic, technical, and administrative standards for health research.

Informed consent was obtained from each participant, documented, and physically recorded. The document recorded the acceptance of the tests, follow-up, and interviews. The project, with its annexes, was approved by the Research Ethics and Methodologies Committee (CEMIN) of the National Institute of Health of Colombia with the code 012/2020.

Additionally, adverse effects were verified by taking the nasopharynx sample. Laboratory results were reported and explained to all study participants.

## 3. Results

The study included a total of 212 workers. Most of them were male workers (73.1%, *n* = 155), mestizos (52.4%, *n* = 95), of medium socioeconomic level 3 (62.4%, *n* = 130), with a technical educational level (39.2%, *n* = 83). The median age was 35.7 years (range 38.3). The majority ethnic group was composed of whites and mestizos (97.5%, *n* = 199). The predominant level of education was university (43%, *n* = 88) [44].

### 3.1. SARS-CoV-2 Incidence, Clinical Presentation, and Risk Factors

In the period between 1 June and 30 August, there were a total of 16 cases of SARS-CoV-2. Most of them were men (*n* = 10), but there were no significant differences (*p* = 0.46). The vast majority were asymptomatic (81.25%, *n* = 13). Only one of the workers developed dyspnea, but he did not require oxygen or other clinical management. The ratio of men and women infected with the virus was 1:1. The accumulated incidence of people with SARS-CoV-2 was 7.92% (95% CI 4.19–11.64). The rate of incidence was 2.62 cases per 100 persons/months (95% CI 1.55–4.17). The hazard function was 7.87 cases per 100 persons/trimester (95% CI 4.66–12.51; Appendix A).

Most of the close contacts were relatives (89%, *n* = 33). In total, 40 RT-PCR tests were performed on 37 family contacts. The proportion of positive contacts was low (16.2%, *n* = 6).

The Poisson regression model found that workers living with partners working from home (teleworking) were 4.5 times less likely to test positive for SARS-CoV-2 (R.R. 0.222, 95% CI 0.064–0.776, *p* = 0.018). Additionally, workers who had longer commutes from home to the office had 1.02 (95% CI 1.002–1.041, *p* = 0.029) times greater risk of having a positive RT-PCR than those who had shorter trips. Variables related to hand-washing frequency and contact with passengers did not have any association with the measured outcomes.

### 3.2. Perception of Risk regarding COVID-19

The perception of individual risk regarding occupational exposure to COVID-19 was classified as medium (mean 5.8 ± 2.6). No significant differences were found in the individual perception of risk between men and women or by socioeconomic stratum (*p* > 0.05). Regarding the risk perception results, there were no differences between the group that tested positive and the group with negative test results (*p* = 0.21).

The response to the question regarding the degree of impact on life due to COVID-19 was medium-high (mean 7.58 ± 2.38). No significant differences were found in the individual perception of risk between men and women or by socioeconomic stratum (*p* > 0.05).

It was found from the question that addressed the perception of the duration of the pandemic that, on average, people consider that COVID-19 will not go away in the medium term ($\bar{x}=$ 6.88 ± 1.78). There were also no significant differences in the response by sex, socioeconomic status, or educational level (*p* > 0.05).

On the other hand, according to the survey, most of the people had not felt symptoms due to COVID-19. The average score of this question was the lowest in the survey ($\bar{x}=$ 6.88 ± 1.78). An association was found between the presence of symptoms related to the coronavirus and sex (x^2^ = 16.34; *p* = 0.038).

The degree of concern about the pandemic situation was at a medium–high level ($\bar{x}=$ 7.38 ± 2.11). No significant differences were found in the individual perception of risk between men and women or by socioeconomic stratum (*p* > 0.05).

Finally, when inquiring about the presence of negative feelings related to COVID-19, it was found that these had a medium frequency in the population $(\bar{x}=$ 5.73 ± 2.47). A negative correlation was found between schooling and the degree of emotional impact by COVID-19 (Spearman’s rho = −0.159, *p* = 0.024; Table 2).

### 3.3. Elements Involved in the Perception of Risk Associated with COVID-19

It was found that those who scored higher in the variables of “how much does COVID-19 affect your life?” and “how concerned are you about COVID-19?” were more likely to have a higher level of self-perceived risk (RR 1.06, 95% CI 1.03–1.10) and (RR 1.06, 95% CI 1.02–1.10; Table 3).

### 3.4. Semistructured Interviews

A total of 10 semistructured interviews were conducted with people who had positive cases of SARS-CoV-2 (six men and four women). All participants had technical or university qualifications. Most of them were married. Seniority on the job ranged from 2 to 14 years (Table 4). Three categories were established, and seven subcategories are presented in Table 5.

### 3.5. Perception of the Risk of Coronavirus Infection

People initially associated SARS-CoV-2 infection with their work activities, such as contact with foreign passengers. This position was more common in workers who had the infection in March. For participants who acquired the infection by July 2020, the perception of risk was more associated with out-of-work time activities such as shopping in supermarkets and taking public transport. By the time the interviews were conducted, the interviewees almost unanimously affirmed that the airport was a safe place. When inquiring about why it is a safe place, they refer to the measures they had adopted, such as the disinfection of common areas, the provision of sinks and gel throughout the building, and the fact that contact with passengers has decreased significantly.
Participant 1: “During the last two weeks of March, we encountered the phenomenon of having several foreign citizens at the airport, either because flights were delayed or flights were canceled, and they were stranded. This represents an inconvenience for the airport in the sense that we do not have hotel capacity, we do not have hotel facilities, so we had to improvise to be able to accommodate people”.
Participant 7: “I feel safe because, in the airport, they disinfect, and I have to go through filters and wash my hands. I feel more insecure being in the street because, at the airport, they disinfect after each shift, so I don’t feel unsafe at the airport”.

### 3.6. Protective Behaviors

The participants declared that current recommendations from the government, such as the use of face masks, frequent hand washing and the use of antibacterial gel, and social distancing, are effective measures for avoiding the transmission of COVID-19. Additionally, it is frequently declared that bathing when arriving from work, avoiding contact with metallic surfaces, disinfecting money and footwear with alcohol, and washing work clothes separately are considered appropriate practices to reduce infection. These practices often came from pieces of advice from friends or articles they had read on social networks (Facebook^®^). Some others mentioned reading articles from traditional media and the Ministry of Health website.
Participant 4: “I have a disinfectant mat at home; when I arrive home, I clean my shoes, wash my hands. I take off all my clothes and put them in the washing machine. Now, I don’t go anywhere without a mask, I have stopped frequenting certain places, I do everything virtually, and as soon as deliveries arrive, I disinfect them”.

### 3.7. Risky Practices Associated with Coronavirus Infection

When inquiring about how workers think they were infected, there was a common idea that the disease caused by SARS-CoV-2 was acquired in the work environment. Some of the workers who were infected by July associated the transmission of the virus with “take away” foods. During lunch time or dinner, it is not unusual to have food delivered. These situations are risky because workers tend to reduce adherence to protocols (hand-washing, social distancing). This situation is also common on weekends at home. Of note, most protocols identify the risk at the workplace, leaving this situation out of the risk matrices.
Participant 9: “I reckon that I was infected because of the roast chicken we order for delivery. We all took off our face masks; we talked and ate at the same time, reaching our hands into the box”.

In the investigation on the areas or activities with the greatest risk for the transmission of the virus, the transit through common areas, such as the food court or the migration area.
Participant 2: “I feel that the immigration arrival area is narrow, and many flights conclude during the peak hours of operation between 7 and 9 at night, with about 15 or 20 flights arriving in normal operation, so I believe that the measures that are going to be taken here in the future are to space precisely these places of arrival, especially because it is congested”.

### 3.8. Recording Symptoms

Some of the participants reported the practice of recording their temperature or making checklists to keep track of their health status. This was used as a mechanism of making sure they were well and had not developed symptoms of the disease to reaffirm their well-being.
Participant 10: “I also took my temperature every four hours. I had my Excel there until I saw that my temperature began to fluctuate between 35 and 37 degrees, I said ok, I’m fine, I stopped there”.

### 3.9. Changing Behavior

The workers also said that COVID-19 has transformed their lives, their relationships with coworkers, and the precautions they take in the workplace to prevent contagion. Due to the experience of having SARS-CoV-2, the workers interviewed have implemented a series of practices in their personal, family, and work lives. Among the individual practices related to their protection and prevention of the spread of contagion are frequent hand-washing and the use of masks. Workers have adopted routines when they get home to prevent infection to their families. It is necessary to mention that none of the participants used a cloth mask, and they referred to masks as the main strategy to prevent contagion. Some of them, particularly in the group who were infected in July, started to avoid food delivery.

On the other hand, the participants stated that they were forced to change some behaviors associated with the confinement situation. Among them are the practice of physical activity and the resumption of family contact that had been neglected before the pandemic.
Participant 5: “I have become tireless with self-care, with hand washing. I carry my gel everywhere; I hardly speak in the workplace. What is necessary, what touches me? It’s a shame because I love to talk, but that is a risk for me and my family, for my colleagues. So, I wash my hands about three times an hour. And when I get home, I take off my shoes at the entrance, disinfect them and go straight for the shower”.
Participant 8: “…I stopped ordering food at home. That’s a lot of risk”.
Participant 4: “I like cycling, practicality. But the people on the street are super relaxed, and there are many people without masks, so neither, I do sports at home, on the roller, I try to support myself, and I don’t take much care of my family”.
Participant 5: “As a result of these feelings that I was going to die, I feel that God gave me another opportunity to live again. Now I talk to my mother almost every day; I thank God that I am alive and I can enjoy her”.

### 3.10. Sharing Experience

After 14 days of isolation and receiving negative test results, people began to generate a series of positive feelings, which resulted in the development of new practices that try to communicate their experience and knowledge to others.
Participant 1: “I downloaded Resolution 666 and put it in an app, where through your cell phone you choose options such as taking a quiz and it is a contest. So, I did that with my family, with the use of masks, hand washing, social distancing, etc., and I started doing it because I felt the need to help”.

### 3.11. Emotions and Feelings in the Face of the Coronavirus

The experienced emotions and feelings during COVID-19 infection were divided into two groups. The first set was associated with the grieving process, passing from surprise to anger and sadness. Fear of dying or infecting loved ones was frequently declared. These negative feelings were reinforced by media information. However, after the critical episode, the participants declared that they were grateful for life and were motivated to carry out altruistic activities such as donating plasma, spending more time with the family, and enjoying more time with their loved ones. In addition, some of them have found ways to share their testimony with others to prevent them from being infected.
Participant 5: “My immediate feeling was death; I felt that I was going to die. That I was going to stop seeing my children (sobs), that I was no longer going to be with my husband. Seeing those things that happened in Spain and Italy, which people died in the street, was impressive. For me, COVID-19 is equal to death”.
Participant 4: “What I feel now individually is wanting to help. I understand that they took a sample from us to determine the existence of antibodies or not, but if I can help you, I will be in the first line to say go ahead, and right now, for me, it is a high priority, to help my family, saying don’t do this or that”.

## 4. Discussion

A mixed sequential explanatory study was conducted on the perception of SARS-CoV-2 infection in workers at the El Dorado, Luis Carlos Galán Sarmiento International Airport in Bogotá, Colombia. The joint result of SARS-CoV-2 risk perception among airport workers is shown in Table 6. To our knowledge, this is the first study carried out in this population that focuses on understanding the risk perception of COVID-19 transmission. We found that the incidence of SARS-CoV-2 infection was 7.9%. The low frequency of the event in this population may be explained by many factors. First, according to the projections of the Colombian Ministry of Health, it was expected that the tendency of the infection curve in Bogotá would reach its peak by September [45]. Second, the early health, safety, and environment policies adopted by the airport in late March were effective in transforming the behavior among workers. In the quantitative analysis, high adherence to practices such as the use of masks and hand washing was shown. Nevertheless, we did not find a significant association between time of training and adherence to practices to avoid infection with the virus. This finding was corroborated during semistructured interviews with workers who had COVID-19. Similar results have been reported by some authors. A study carried out in a population of healthcare workers found that the level of knowledge of COVID-19 was associated with high adherence to face mask usage and frequent hand-washing [46]. In another study carried out in Malaysia, the authors reported that people who had more knowledge tended to report more adherence to hand-washing [47].

Our study reported that perception of risk for COVID-19 was classified as a medium–high level. In the regression model used, it was found that the questions on the concern and impact on life due to the pandemic had a higher self-perceived level of risk. However, when incorporating the elements of the interviews, it was found that El Dorado was one of the first places where the COVID-19 emergency was noticed due to the modifications that they had to develop to serve a large number of passengers who were waiting for flights and had to remain in the terminal facilities for several days. In this sense, the population of workers was forced to collectively adopt early protective behaviors such as hand washing, use of antibacterial gel, and use of face masks because they associated the potential interaction with passengers coming from abroad as a risky activity.

However, it should be mentioned that adherence to the recommendations may be influenced by the conditions in the environment, such as the availability of protective equipment, soap, and hand sanitizer gel [48,49]. The workplace conditions that are related to the availability of elements for hand hygiene may change between workers who go to the office five days a week and those who spend three days a week there. We did not observe significant differences among the incidence of SARS-CoV-2 transmission between workers in different work modalities. However, the transmission of the virus was associated with longer trips from home to office, independent of the mode of transport. This issue was controversial, considering that many authors have indicated the role of public transport as a potential vector of transmission for SARS-CoV-2 [50].

Participants did not identify any activity or job role that was associated with the outcome of having the SARS-CoV-2 infection. However, in the interview, some transit areas emerged that should be considered in a future normalization of operations, such as migration areas and food courts. In terms of the participants, these areas are associated with a considerable number of national and foreign people. Additionally, some risky daily activities related to lunch or dinner time were reported, where the recommendations of social distancing are easily avoided. This is naturally understandable, considering that the act of eating is more than a biological necessity; it is also a social activity. The interaction during meals, even during work time, should be addressed to reduce the negative impact of the pandemic on eating behaviors. Some authors have shown the negative outcomes of social distancing on eating disorders, with a high frequency of depressive and anxiety symptoms [51].

The finding of a moderate-to-high level of risk perception coincides with that reported in the literature. A study that reported the level of risk perception of COVID-19 in the economically active population of the countries of Europe, North America, and Asia found that the perception of risk of SARS-CoV-2 is uniformly high without being affected by elements such as ethnicity, socioeconomic status, or gender [52]. However, other studies conducted earlier in the pandemic have different results. A study carried out in the United States during the first months of the pandemic reported that adherence to recommendations to prevent infection was low, mainly motivated by optimism bias in the surveyed population [53]. In our case, it is possible that this bias did not exist because at the time the survey was applied, the country had already been in an emergency for several months, and the airport had ceased operation.

It should be noted that the quantitative analysis showed no association between the feelings caused by the pandemic and risk perception. However, in the interviews, it emerged that the emotions generated by the pandemic are frequently negative, associated with fear and risk of death, but participants also felt the concern and desire to help others through their testimony after going through the episode of infection. These findings coincide with what was reported in a study on the perception of risk of COVID-19 carried out in Italy [54]. The authors found that risk perception was mainly linked to factors related to the quality of life but was also strongly influenced by other people’s emotional concerns [54].

These findings are important for the formulation of public policies aimed at increasing adherence to measures to contain the pandemic, which allows a better understanding of why some workers maintain the recommendations for the use of personal protection items and hand washing for longer. In this way, this work illustrates how the perception of risk towards COVID-19 is mediated by experimental and sociocultural factors, understanding that in this way, the risk of transmission of COVID-19 exists but is permeated by elements such as work conditions, experience, and social background [10].

The study had several limitations. First, there are those related to the type of study design; the central objective of a longitudinal study is to know the health status of a specific population in a defined time and place [55]. However, the main difficulty of this study stemmed from the fact that the virus was already present in Bogota by June 2020. In this sense, it was not possible to differentiate the exposed group from the nonexposed group. We tried to reduce the risk of a selection bias by using RT-PCR and serological antibody tests at the enrollment. However, the serological tests have inherent limitations related to antibody identification. Second, during the quantitative analysis, we observed no statistical differences related to practices, risk perception, and usage of personal protective equipment between workers who had SARS-CoV-2 versus workers without the infection. Therefore, we did not include workers with negative RT-PCR results in the interviews. Third, the instrument used to quantify the perception of risk was based on a Spanish version of the revised Illness Perception Questionnaire. Despite the fact that the instrument was validated with Castilian speakers, it should be validated in the Latin-American context. In the qualitative analysis of the data, it was not possible to interview all study participants who had a positive RT-PCR, and the saturation criterion was not reached, which is why the analysis remained at a descriptive level. Moreover, the analysis of the transcripts was carried out by only one researcher. Nevertheless, the three coauthors validated the codes and categories from the transcription. We tried to reduce this bias by the peer debriefing process. Finally, these are the aspects related to the conditions in which the sample is taken and the ability of RT-PCR to identify viral RNA in the first two days of infection. These aspects have been considered and will be reduced as much as possible by training the research team in taking, packing, and processing the samples. As a quality control, some samples were sent to a reference laboratory at the Charité Institute of the Universitätsmedizin in Berlin [56].

## 5. Conclusions

The SARS-CoV-2 virus has modified life and working conditions worldwide. This represents a challenge for public health and occupational health. Here, we conducted a mixed-methods research study of airport workers during the SARS-CoV-2 pandemic to understand the dynamics of the infection and risk perception of SARS-CoV-2 in a cohort of workers at El Dorado airport, Luis Carlos Galán. Workers who declared greater affection for life and showed more concern about the pandemic were more likely to have a higher level of self-perceived risk. A high perception of risk was associated with activities such as shopping in supermarkets and using public transport. These considerations were concordant with the quantitative data that showed a higher risk of transmission of the virus on longer trips from home to office. Risk perception is strongly influenced by practices related to work conditions and the environment. The adoption of measurements such as availability and usage of protective equipment and the frequent disinfection of potentially contaminated areas is associated with low risk perception. Surprisingly, the level of risk perception was not associated with COVID-19 diagnosis.

## Figures and Tables

**Figure 1 ijerph-17-09002-f001:**
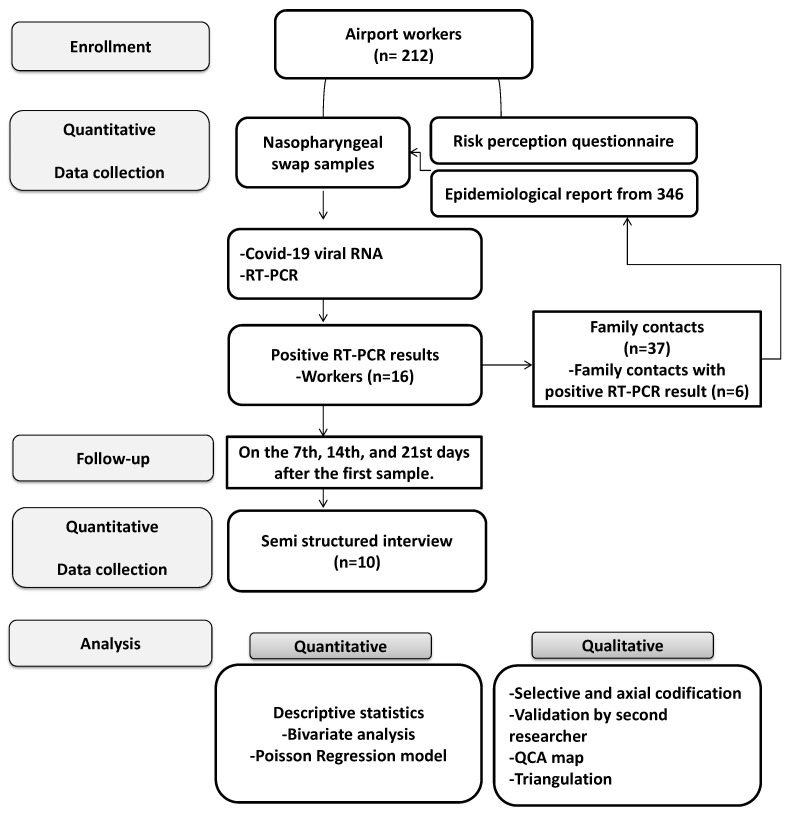
Description of the research process in the study.

**Table 1 ijerph-17-09002-t001:** Pre-established categories, definitions, and questions of the interview guide.

Category	Definition	Question
Perception of the risk of the contagion of coronavirus in the workplace and at home [11]	Individual perception about the probability of the infection occurring in the workplace or of infecting others	-What do you know about COVID-19? -Which positions do you consider to be the highest risk for contracting coronavirus? -Are there any areas of the airport that you particularly think are more likely to infect workers?
Intra and extraoccupational risk practices related to the spread of SARS-CoV-2 [30]	Practices are material events that constitute social reality. The practices are linked to beliefs, emotions, and conceptions	-What practices have you implemented as a result of the coronavirus infection? -What practices do you consider risky in your work? -What practices have you implemented in your home and your workplace?
Emotions about the coronavirus [31]	It is the set of feelings, hunches, and beliefs around a particular situation	-What feelings does the coronavirus generate in you? -How do these feelings influence the way you do your job?

**Table 2 ijerph-17-09002-t002:** Airport workers’ perception of risk regarding COVID-19.

Question	Mean	Median	SD	Range
How exposed do you feel to the coronavirus in your job?	5.88	6	2.61	9
How much does coronavirus infection affect your life?	7.59	8	2.38	9
How long do you think the coronavirus infection will last?	6.88	7	1.79	8
Do you feel symptoms due to coronavirus infection?	2.00	1	1.78	9
How concerned are you about coronavirus infection?	7.39	8	2.11	9
How emotionally does the coronavirus infection affect you?	5.74	6	2.47	9

**Table 3 ijerph-17-09002-t003:** Poisson regression model for risk perception of COVID-19 (RPQ) among airport workers.

Variable	Crude RR (95% CI)	Adjusted RR (95% CI)	*p*
Age	1.00 (0.99–1.01)	1.001 (0.99–1.00)	0.99
Sex (female)	0.89 (0.78–1.01)	0.92 (0.81–1.06)	0.27
Socioeconomic level	0.99 (0.93–1.06)	1.01 (0.94–1.09)	0.99
Risk level (low)	1.24 (1.01–1.51)	1.27 (1.04–1.55)	0.19
Ethnic	1.14 (1.03–1.26)	1.08 (0.97–1.20)	0.13
Not having had symptoms of COVID-19/RT-PCR (−)	1.03 (1.00–1.06)	1.00 (0.97–1.12)	0.64
Having had symptoms of COVID-19/RT-PCR (+)	1.02 (0.93–1.12)	1.01 (0.92–1.12)	0.73
Effect on life due to the pandemic	1.091 (1.06–1.12)	1.06 (1.03–1.10)	0.00
Concern about the pandemic situation	1.09 (1.06–1.12)	1.06 (1.02–1.10)	0.00
Emotional impact by the pandemic	1.04 (1.02–1.07)	0.99 (0.96–1.02)	0.56

**Table 4 ijerph-17-09002-t004:** Sociodemographic characteristics of the interview participants.

Worker Roll	Age Years	Sex	Educational Level	Marital Status	Work Experience	COVID-19 Diagnosis Date	# Days after RT-PCR Was Negative	Disease Course
Administrative	52	Male	Postgraduate	Married	14	23/03/2020	7	Asymptomatic
Administrative	43	Male	Postgraduate	Married	8	23/03/2020	21	Asymptomatic
Administrative	44	Male	Postgraduate	Married	5	23/03/2020	14	Mild disease
Operative	38	Female	University	Married	6	7/07/2020	21	Mild disease
Operative	29	Female	University	Single	2	7/07/2020	21	Asymptomatic
Administrative	31	Female	Postgraduate	Married	4	7/07/2020	14	Asymptomatic
Operative	35	Male	University	Married	4	23/03/2020	7	Asymptomatic
Operative	38	Male	Technical	Married	5	7/07/2020	14	Mild disease
Administrative	35	Female	University	Single	4	7/07/2020	21	Mild disease
Operative	40	Male	University	Married	8	7/07/2020	14	Mild disease

**Table 5 ijerph-17-09002-t005:** Emergent categories and subcategories for qualitative analysis.

Category	Subcategory	Definition
Practices	Protective behaviors	Activities or behaviors that decrease the transmission of SARS-CoV-2
	Risky practices associated with coronavirus infection	Activities or behaviors that favor the transmission of SARS-CoV-2
	Recording symptoms	Activity to get a record of the symptoms experienced
	Changing behavior	Set of activities carried out by people aimed at modifying their behavior in the face of a specific experience
	Sharing experience	Manifest activity by the worker where he or she declares the need to publicize their experience of the SARS-CoV-2 infection
Emotions in the face of the coronavirus	Experienced feelings during the COVID-19 episode	Feelings or emotions expressed by people who have had COVID-19
	Post-COVID-19 feelings and emotions	Feelings or emotions expressed by people who have had COVID-19

**Table 6 ijerph-17-09002-t006:** Joint result of SARS-CoV-2 risk perception among the airport workers.

Quantitative Results		Qualitative Results			Mixed Methods Inference
Instrument	Findings	Category	Subcategory	Findings	
RT-PCR for SARS-CoV-2	Accumulate incidence (7.5%)	Practices	Protective behaviors	Frequent hand-washing, after work showers, shoe disinfection was considered protective practices	Adherence to the recommendations may be influenced by the conditions in the environment, such as the availability of protective equipment, soap, and gel. Moreover, the promotion of protective behaviors should involve people with whom the worker lives.
	Asymptomatic cases (81.25%)		Risky practices associated with the contagion of coronavirus	Risky practices are associated with keeping in touch with foreign passengers and activities out of the work (visiting shopping centers, supermarkets, banks). The risk of transmission is associated with public spaces	Recommendations to prevent the transmission of SARS-CoV-2 should not be limited to the work area. It should include the extra-work sphere.
	Positive close contacts (16.2%)		Changing behavior	Increasing physical activity, avoiding crowded places, preventing using public transport and touching metallic surfaces	The transmission of the virus was associated with longer trips from home to office, independently from the mode of transport. Participants who experienced COVID-19 considered the usage of public transport as a risky practice.
	No outbreaks per area reported during the period		Recording symptoms	Recording the experienced symptoms during the COVID-19 is a practice used to check the well-being	
Risk factors and sociodemographic characterization	Prolonged trips from home-office increased risk		Sharing experience	Telling the experience of COVID-19 with relatives and coworkers was a practice declared by participants who had the disease.	Expressing emotions during and after the COVID-19 episode may be an opportunity to reinforce the surveillance system and communicate the risk in the workspace
	Workers living with a person working at home reduces the risk of infection	Emotions in the face of the coronavirus	Experimented feelings during the COVID-19 episode	Fear of death, anger, anguish, uncertainty are the feelings associated with the COVID-19	
	High adherence to the usage of face mask and frequent hand washing (98%)		Post-COVID-19 feelings and emotions	Appreciation and the feeling of having a new opportunity.	
COVID-19 risk perception questionnaire	Risk perception medium–high	Risk perception	Risk perception	The perception of risk is medium to high. Nevertheless, the job place is perceived as a safe place to work	The promotion of visible individual protective practices such as frequent hand washing and wearing facemask was associated with the risk of transmission of SARS-CoV-2
	Risk of transmission is associated with activities where passengers are involved				

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
