# Peer review of "Infection and Risk Perception of SARS-CoV-2 among Airport Workers: A Mixed Methods Study"

_ijerph, 2020, doi:10.3390/ijerph17239002_

Round 1
Reviewer 1 Report
The manuscript by Malagon-Rojas et al describes a mixed-method study analyzing the infect and risk perception of SARS-CoV in airport works at the El Dorado airport in Bogota Columbia. The authors used a PCR based screening system to identify cases of COVID-19 and then followed up with interview and questionnaire to assess their attitudes and behaviors around the pandemic. The paper is well written, and the data support the conclusions, despite a low number of positives making more quantitative analyses not possible. Despite this, the authors showed that workers with partners working from home were at a decreased risk of infection, while workers with longer commute times had increased risk of infection. The qualitative findings are intriguing, and the authors do a good job showing restraint and do not overinterpret their results. The methods section is very well detailed, making it easy for others to repeat or expand upon the work.
Recommended revisions:
- In section 3.1 the authors mention the “proportion of positive family contacts was 16% (n = 1)”. Of the 13 workers who were positive, how many did the authors test the family members/contacts? How many family members/contacts did the authors test in total? Please provide additional information about what and how sampling occurred in the family and friends of the infected workers.
- Please provide additional context for the protective behaviors in section 3.6. A number of these were not universally recommended or adopted. Why did the authors choose to discuss these protective behaviors? The ones recommended from the government are more widely adopted, but “avoiding contact with metallic surfaces” for example was not. Where did this second set of recommendations come from?
- Were any interviews performed on individuals that did not contract COVID-19? If so, how were they chosen? A comparison of the behaviors and attitudes of those who caught COVID-19 versus those who did not could further support the conclusions.

Reviewer 2 Report
This is an interesting study addressing the COVID-19 pandemic in a key population. The mixed methods approach is really insightful for answering the study question. Below are some comments for improvement: - In line 63-64, the authors indicate that the quantitative component of the study was a prospective cohort study. However, the authors do not provide details on the length of follow-up of participants. The quantitative aspect seems like a cross-sectional study given the information provided so the authors should clarify the methodology. - In lines 77 to 80, the authors describe their sample size calculation. Given the variety of professions included in the group of airport workers, it is likely that they have different exposure risks. -In table 3, line 236, the authors report results of Poisson regression in which they include qualitative variables ,(e.g. perception of risk by COVID-19). Regression analysis is usually used for quantitative variables and qualitative variables have a different analysis approach. - If variables such as "perception of risk" are converted to a numerical variable with a score, then the authors should explain this scoring in some detail along with an interpretation of the scores. This would enable the reader to better understand the meaning of the results. -The authors report figures of incidence of SARS-CoV-2 in the results section. However, the reported measure of effect seems to be a proportion. In line 450, the authors indicate that they are using a cross-sectional design. Please clarify what effect measure was used and amend the result accordingly.Author Response
Please see the attachment

Reviewer 3 Report
This is a very timely and important research amid the ongoing COVID-19 pandemic. Overall quality of the paper is satisfactory. However, there are rooms for improvement.
- I suggest the paper be sent for English proof-reading as there are inconsistencies and minor errors throughout the text. The readability of the manuscript is not very high despite the importance of the study. For example, COVID-19 and Covid-19 were both used throughout the text. In addition, the usage of SARS-CoV2 and COVID-19 are inconsistent and confusing. Please make sure that SARS-CoV2 is used to refer to the virus and COVID-19 is used to refer to the disease.
- Title: Instead of mixed method, the right term is mixed methods.
- Lines 62 to 67: The explanation is very vague. If you are using STROBE and SRQR, it is better to include the checklists as attachments unless instructed otherwise by the journal.
- Lines 136-153: In what language was the interview conducted? How were the independent variables selected in the final Poisson regression model?
- For mixed methods, both quantitative and qualitative data should be integrated to generate new insights not achievable by either data set alone. I understand that the authors integrated the insights in the discussion. However, this might not be easy for readers to follow so if possible, I would suggest the authors to create a joint display to combine and contrast both datasets in joint displays.
- The presentation of the qualitative results could be simplified shortened. It is a little difficult to read. In addition, the discussion could also be simplified and focus only on the important points worth of policy recommendations.
Reviewer 4 Report
This paper conducted a mixed approach to describe the infection and risk perception of SARS-COV-2 in airport workers. There are some concerns that should be addressed.
- Figure 1 should be redrawn. The capitalization/lowercase, the arrows should follow a consistent logic.
- The major study is very descriptive. Any relationships? If not, it is more like a report rather than an academic paper.
Reviewer 5 Report
Please see attached file for my comments. Thank you.

Round 2
Reviewer 2 Report
The authors have addressed all the feedback regarding the methodology.
The manuscript needs copy-editing for language prior to publication.
